# Ensemble Models for Tick Vectors: Standard Surveys Compared with Convenience Samples

**DOI:** 10.3390/diseases10020032

**Published:** 2022-06-08

**Authors:** William H. Kessler, Carrie De Jesus, Samantha M. Wisely, Gregory E. Glass

**Affiliations:** 1Emerging Pathogens Institute, University of Florida, Gainesville, FL 32611, USA; willhkessler@gmail.com (W.H.K.); wisely@ufl.edu (S.M.W.); 2Geography Department, University of Florida, Gainesville, FL 32611, USA; 3Department of Wildlife Ecology and Conservation, University of Florida, Gainesville, FL 32611, USA; carriedejesus@ufl.edu

**Keywords:** ensemble models, Species Distribution Models SDMs, ticks, *Amblyomma americanum*, *Ixodes scapularis*, Florida, biased sampling, study design

## Abstract

Ensembles of Species Distribution Models (SDMs) represent the geographic ranges of pathogen vectors by combining alternative analytical approaches and merging information on vector occurrences with more extensive environmental data. Biased collection data impact SDMs, regardless of the target species, but no studies have compared the differences in the distributions predicted by the ensemble models when different sampling frameworks are used for the same species. We compared Ensemble SDMs for two important Ixodid tick vectors, *Amblyomma americanum* and *Ixodes scapularis* in mainland Florida, USA, when inputs were either convenience samples of ticks, or collections obtained using the standard protocols promulgated by the U.S. Centers for Disease Control and Prevention. The Ensemble SDMs for the convenience samples and standard surveys showed only a slight agreement (Kappa = 0.060, *A. americanum*; 0.053, I. scapularis). Convenience sample SDMs indicated *A. americanum* and *I. scapularis* should be absent from nearly one third (34.5% and 30.9%, respectively) of the state where standard surveys predicted the highest likelihood of occurrence. Ensemble models from standard surveys predicted 81.4% and 72.5% (*A. americanum* and *I. scapularis*) of convenience sample sites. Omission errors by standard survey SDMs of the convenience collections were associated almost exclusively with either adjacency to at least one SDM, or errors in geocoding algorithms that failed to correctly locate geographic locations of convenience samples. These errors emphasize commonly overlooked needs to explicitly evaluate and improve data quality for arthropod survey data that are applied to spatial models.

## 1. Introduction

Growing concerns about ticks and tick-borne diseases (TBDs) led the U.S. Centers for Disease Control and Prevention (CDC) to refocus attention on better identifying the geographic distributions of vectors, elucidating factors associated with risks and targeting interventions [1,2]. One of CDC’s major contributions was developing standard survey protocols to improve the quality of information on the occurrence and geographic ranges of various hard bodied, ixodid, tick species within North America [3,4].

The standard survey protocols [4] (pp. 9–14) are designed to address increasingly more explicit information regarding geographic ranges and details of the timing, locations and extents of risk for TBDs. Specific survey methods limit what questions can be resolved with various approaches. Although there are numerous strategies to identify the presence of ticks (such as removal from people or animals, CO_2_ traps, or flagging) more critical questions related to vector density and prevalence of infection are restricted to only a few study designs [4] (Table 2). One of the major aims of these protocols is to estimate vector distributions. When this task uses geopolitical boundaries, such as countries or smaller geopolitical subunits the products can be represented by choropleth maps in which political regions may be shaded based on a defined scheme. However, environmental conditions that impact vectors rarely correspond to these boundaries. Consequently, approaches have been developed to combine information on vector occurrences with extensive environmental information to produce expected ranges of ticks throughout a geographic region. These Species Distribution Models (SDMs) differ in many of their assumptions and can be aggregated into ensemble SDMs. However, SDMs incorporate additional restrictions into the types of data that can be applied to produce accurate results.

Although improved results for SDMs would be expected from applying standard surveillance methods (and incongruities more rapidly detected), there are also challenges in implementing standard surveys. First, because these are new protocols most data will be prospectively acquired, so having enough staff trained in the protocols requires time, coordination among various agencies and financial resources to conduct prospective surveys [5]. Second, altering traditional surveillance protocols may leave some historical information unusable in the absence of post hoc modifications that control for biases in the original studies [6]. Third, it is unclear if the new protocols generate results that are both internally valid and produce substantially different results than passively collected data sources that make the incurred costs worthwhile. Glass and colleagues [7] showed that the predicted distribution ranges of three ixodid tick species in mainland Florida that were generated from the CDC protocols had high levels of internal consistency and external validity when new sites were sampled in subsequent years. However, they left open the question of how different the results generated from new survey methods were compared to those from previously collected data.

An ad hoc solution in the absence of new survey data has been to use previously obtained data that may have been gathered for any purpose. These data may be represented by preserved specimens or reports of collected ticks. These records are considered as convenience samples as they have no coherent design in their collection and are gathered in any number of ways, but are more easily available for input into SDMs as vector location information. However, historically, convenience samples also have limitations. Generally, convenience sampling introduce biases and conflates where ticks occur with where the sampling was performed [6,8]. If sampling is biased by human activity or host animal habitat selection, these regions may be overrepresented while other habitats may be under sampled—altering the relationships between vector species and environmental factors. Another, less obvious challenge occurs in identifying the geographic locations of historical samples, themselves. Traditionally, location data are often recorded as a position relative to a geographic place name (street address, city, distance from an identifiable geographic feature). Most geocoding software have algorithms built into the systems that convert the place names into geographic coordinates. When these are input for analyses into SDMs, the spatial errors associated with assigning the coordinates may be large relative to the scale of the analysis. Jacquez [9] noted that these errors, themselves, have marked spatial heterogeneity (rural sites tend to have significantly larger spatial errors than urban or suburban locations) making the assumption of stationarity in the errors invalid, across space. Few studies explicitly report these errors prior to analysis or model construction [9]. Prospective studies permit sampling locations to be identified at high resolution, spatially, either locating samples with large scale maps, aerial photography, or global positioning system (GPS) receivers (or similar systems) [10]. Retrospective surveys are limited to the implicit scale of the original recorded data [6,9]. If there is discordance between this scale and that of the environmental data, misclassification is likely. These spatial errors are related to the algorithms that convert a place description to geographic coordinates (geocoding) [9,11,12] but the consequences for modeling SDMs appeared have been ignored in the applications literature. This issue has been most frequently evaluated when a street address is the key field in geocoding efforts [12,13].

To consider these challenges, teams from the Southeastern Regional Center of Excellence in Vector Borne Diseases (SECVBD; https://cdcsercoevbd-flgateway.org/ accessed 7 March 2022) performed surveys using the CDC protocols (standard surveys) to generate predicted distributions of the two most commonly collected, human-biting ixodid tick species in mainland Florida [7,14,15]. These species *Amblyomma americanum* (Linnaeus, Ixodida, Ixodidae) and, *Ixodes scapularis* (Say, Ixodida, Ixodidae) are often implicated as vectors of viral and bacterial pathogens in Eastern North America [1,16].

An earlier study of the standard survey methods [14,17] showed that standard sampling protocols generated ensemble models of SDMs that were consistent. By consistent, we mean that validation surveys at new locations in different years produced survey results with very high (91–100%) sensitivity and increasing specificity as SDM concordance increased for all the tick species, consistent with the original ensemble SDMs [7]. The ensemble SDMs had the added advantage that they narrowed the geographic extents of the species ranges within Florida by 25–50% [14,17], suggesting risk for TBDs in Florida was much more restricted than previously presumed.

Given the advantages and disadvantages, in time, resources and biases, when using standard surveys or convenience sampling, it may be useful to determine if the different frameworks generate meaningful, practical, differences in outcomes that would impact their applications in surveillance and intervention [3,4]. To determine how different the results of predictions might be, we extended our studies using two frequent sources of convenience samples to build ensemble models for *A. americanum* and *I. scapularis.* We compared these ensemble models with the ensemble models from the standard surveys [14]. The convenience samples were obtained from a natural history museum and from an administrative department in the State of Florida that maintained databases of collected ticks for citizens and scientists during past decades. We then attempted to identify reasons for discrepancies in the resulting ensemble models [9].

## 2. Materials and Methods

Field Specimens: Protocols for standard tick surveys performed in mainland Florida from 2015–2019 are described in detail elsewhere [7,17]. Briefly, regions of the state were subdivided into six climatic regions identified by NOAA [18] and 5 category I land cover classes [19]. Sites were selected within these regions in proportion to those categories. Within sites, pairs of approximately 150 m transects were flagged bimonthly for questing ixodid ticks. At least one pair of transects were performed for each major landcover class at each site. Collected ticks were removed from flags and stored in ethanol for each transect. Ticks were identified using morphological keys with a dissecting microscope [20].

Convenience samples were obtained from two sources common to retrospective sampling for SDMs, museum collections and public survey materials. The museum samples were obtained from the University of Florida Museum of Natural History (FLMNH). Immature ixodid ticks were identified on various lizard species collected within Florida [15]. Individual lizards were examined under a microscope for larval and nymphal ticks that were removed and identified based on morphological features and associated with the collected vertebrate [15,20]. The collection location was extracted from the UF herpetology digital database of specimen records (http://specifyportal.flmnh.ufl.edu/herps/ accessed 7 March 2022) and converted to the nominal latitude and longitude using the resident geocoder.us option.

Public survey records were provided by the Division of Plant Industry within the Florida Department of Agriculture and Consumer Services [21]. These records included year of collection, county, town and further geographic indicator (such as street address or other geographic locator) and tick species name. The specimens came from various sources, including the public, and wildlife surveys. Locations were geocoded interactively using ArcGIS ver 3.2 and TIGR base maps (geocoding basis) by one of the authors (WK).

Analyses: To create SDMs for the convenience samples, background locations were obtained by random sampling until there were 10 times the numbers of locations for each of the species in the convenience samples [14]. Any background site that included a site where a tick had been identified was excluded and a new random location was drawn. For standard surveys, transects that never yielded a tick species were used as ‘absence’ locations [14].

To compare the ensemble model results from the convenience and standard data, five SDMs were used to generate ensemble models for *Amblyomma americanum* and *Ixodes scapularis*. Details of models, environmental predictors, and variable selection criteria followed Kessler et al. [14,22]. Environmental predictors used for the SDMs included 36 characteristics of the climatic and habitat conditions for mainland Florida. Habitat variables used three descriptions of Normalized Difference Vegetation Index (NDVI; minimum, mean, and maximum), land cover, elevation, soil, geomorphologic characteristics, and distance from surface water features. The climatic characteristics were 19 measures of temperature and precipitation variability.

Variables describing NDVI, elevation, soil, and geomorphologic characteristics were derived from MODIS NDVI 16-day composites, ASTER Global DEM, STATSGO soils databases and national hydrography datasets, respectively [23,24,25]. Land cover used the Florida Cooperative Land Cover database—a statewide classification of all major land cover types, from 2008–2011, at a native resolution of 10 m [19]. This is a hierarchical classification. The primary state-level land cover types were aggregated using a majority rule to the coarsest level of five primary types: forest, including pine and hardwoods; shrub, encompassing shrub and brush lands; grasslands/pastures; wetlands; and a final, general category including all other land types such as water bodies, urban areas, and seasonal agriculture. All environmental predictors were resampled using bilinear interpolation for continuous variables and majority rule for categorical variables and cropped to the same extent with 1 ha resolution. The climatic variables consisted of 19 bioclimatic variables calculated from gridded daily temperature and precipitation estimates in Daymet [26]. The climatic variables were calculated and used the naming conventions of Hijmans et al. [27] as implemented in the ‘biovars’ function in the dismo R package [28,29]. The bioclimatic variables were calculated at the native 1 km^2^ resolution of the gridded daily estimates.

SDM development included an initial environmental variable evaluation. The potential variable set was reduced following Springer et al. [29]. We, then, applied five modeling algorithms, using the dismo R package [28]: (1) general linear models (logistic regression—LR) [30], (2) boosted regression trees (BRTs) [31], (3) random forests (RF) [32], (4) multivariate adaptive regression splines (MARS) [33], and (5) maximum entropy (MaxEnt) [34], to estimate the distribution of each species. Each method is considered to perform well using presence/absence or presence/background data [35,36]. MaxEnt was supplied with the same absence data as the other algorithms for the standard surveys and the same background data for the convenience samples. Optimization criteria for each model are described by Kessler et al. [14]. Validation of the final models used a 10-fold cross validation to assess the agreement between observed and predicted values. Each model was trained by withholding one-fold of the data and generating the model with the remaining folds. The withheld fold was used to test the predictive accuracy of each run. This procedure was repeated ten times so that each fold was withheld and used for testing. All models were validated using the same folds for consistency across methods.

The geographic ranges of estimated suitability for each species applied the optimized algorithms to the rasterized environmental variables at 1 ha resolution. The output was a probability surface for the entire mainland portion of the state, representing the probability that the conditions in each raster cell were suitable for each species. These probability surfaces were dichotomized as suitable or unsuitable based on a threshold criterion for prediction accuracy (sensitivity = specificity) [14,22]. Thus, suitability was based on a consensus score ranging from 0–5 ranking how many of the algorithms indicated whether a given hectare was suitable or not.

Metrics: During standard surveys, a transect was scored as ‘positive’ if the species was found on it during any survey and as ‘negative’ if ticks was never sampled during any survey. The ensemble predicted ‘present’ if any of the five models predicted occurrence for the tick species at the transect and ‘absent’ if none of the five models predicted occurrence at the transect.

All summary areal measures of the ensemble models for standard and convenience sampling were measured in hectares and converted to square kilometers. The numbers of transects yielding *A. americanum* or *I. scapularis* in each of the six (0–5) model categories were cross tabulated by survey method. To compare the ensemble maps from standard and convenience sampling, Kappa measures [37] were used to evaluate the predicted geographic overlap on a pixel-by-pixel basis. Standard errors and 95% confidence intervals were calculated according to Fleiss et al. [38]. Weighted Kappa was used because pixels that were scored similarly, but not exactly, by the convenience and standard survey methods would convey important confirmatory information. For example, if one map scored a pixel as 0 and the other map scored it as 1, this suggested that the two survey methods produced more similar results than if one map scored a pixel as 0 and the other scored it as 4. We used published thresholds [39] to assess the Kappa, where <0 indicated no agreement, values between 0 and 0.20 slight agreement, 0.21–0.40 fair agreement and >0.40 moderate or better agreement.

Transects where a tick species was found but no models predicted presence were considered errors of omission for the ensemble [7]. Distances to the centers of the nearest pixels with a predicted occurrence by SDMs were calculated in ArcGIS. Locations of the convenience samples that were preliminarily identified as errors of omission by the standard survey maps were physically revisited, based on the reported longitude-latitude, to identify possible sources of errors and to evaluate potential causes of positional errors, according to Jacquez [9]. To examine the association between the year of collection and the proportion of misclassified convenience samples a 4 × 2 Association test was performed, with three degrees of freedom. The preliminary numbers of ‘error of omission’ sites for the intervals prior to 1940, 1940–1959, 1960–1979, and ≥1980 were examined for convenience sites falling outside standard SDM pixels. Effect of data source (museum species vs. citizen science) was dichotomized and whether those convenience samples were errors of omission was examined for the standard survey SDMs after 1979, (when the two data sources had the greatest temporal overlap).

## 3. Results

Collection results: There were 560 transects surveyed for adult ticks using the standard collection protocols. These transects were located at 41 sites [7,14]. Of all the transects, 98 (17.5%) yielded *A. americanum* and 65 (11.6%) yielded *I. scapularis*. Details of sampling results were previously reported [7,14]. The geographic locations of standard survey sites and their classification (presence/absence) are shown in the Appendix A. In comparison, the convenience surveys reported 53 unique locations that yielded *A. americanum* (10 from reptiles at FNHM and the remainder reported to the FL State department) and 178 locations with *I. scapularis* (133 from reptiles at FMNH and 45 from the FL State department). Their locations are shown in the Appendix A. The museum collections were from reptiles collected from prior to 1940 through 1999. The majority were collected between 1960–1979. In contrast the FL State Department specimens were collected as early as 1980 and as late as 2017. The majority of these records were between 1980–1999. Notably, 12 sites in southern mainland Florida yielded *I. scapularis*. These ticks were, predominantly (9/12 sites), immatures collected from lizards [15], These regions were anticipated by standard ensemble models [7,14] but flagging did not yield adult ticks during the four years of surveys [7,17].

SDMs: The SDMs for both species identified different suites of environmental variables, depending on the species and whether standard surveys or convenience sampling were performed (Table 1). Performance metrics for standard surveys are previously described in [17] (Table 4). The performance metrics for the convenience samples (Appendix A) were generally comparable to the standard survey metrics. The accuracy, threshold and AUC standard errors did not show a significant difference from the standard models. However, the AUCs for the convenience survey models tended to be lower than the AUCs of the standard survey models.

For both survey methods BRT, RF and to a lesser extent MARS, identified the same suite of environmental variables, while LR and MaxEnt identified many of the same variables but these differed from the other three models (Table 1). RF and BRT were robust to survey method—identifying many of the same environmental predictors whether ticks were collected by convenience or standard surveys. Among climatic variables, isothermality, the extent to which daily temperature range was similar to the annual temperature range (bio3), was most commonly associated with tick occurrence—regardless of species or method. Otherwise, various measures of precipitation, especially its seasonality, (≥bio12) were more frequently associated with distinguishing tick occurrence, than were temperature variables. Elevation (dem), steepness of the slope (curve) and distance to nearest surface water were also commonly associated with tick occurrence. BRT and RF incorporated most of the land use/land cover classes in their classifications but these were less frequently identified by the other SDMs.

Comparing standard and convenience SDMs: For both standard surveys and convenience sampling, collections of ticks were more likely as agreement among the SDMs increased, regardless of species (Figure 1). Standard surveys were more likely to yield ticks than the convenience samples when there was a concordance among the SDMs (3–5 SDM’s). For example, when all five SDM’s generated from the standard surveys identified a region as having *I. scapularis*, 70.3% of the transects yielded the ticks, while only 33.9% of the sites from the convenience sampling represented positive sites. Similarly, 78.6% of standard survey sites yielded *A. americanum* when all five SDM’s agreed but only 52.8% of the convenience sampling were positive for the species.

The estimated extent of the mainland with ticks varied by species and collection strategy (Table 2). Generally, however, both survey methods excluded ticks in approximately 40—45% of the mainland. Areas of concordance (three or more SDMs classified ticks as present) ranged from nearly 20% (convenience samples of *A. americanum*) to 30% (convenience samples of *I. scapularis*).

Overall, the ensemble maps for standard and convenience surveys were broadly similar within the mainland (Figure 2). Both methods generated ensembles indicating more of the SDMs anticipated ticks in the northern half, rather than the southern half of the state. However, the standard surveys generated a greater geographic concordance among SDMs (yellow, orange or red) in the northeast and north central portion of the mainland while the convenience survey ensemble model had its greatest concordance shifted towards the central and western portion of north Florida. Consequently, the convenience sampling ensemble models for both species indicated reduced agreement of tick occurrences to the east of the St John’s River (Figure 2). Additionally, convenience data suggested a broader, more diffuse, region (greens and yellow) of occurrence in the southwestern portion of the state from Tampa Bay southward towards Lake Okeechobee. In contrast, standard survey maps indicated that both species occurred southwards from Lake Okeechobee towards Fort Myers. The standard surveys indicated ticks were absent from most of the Everglades while the convenience samples indicated a more extensive region of absence along the southern and eastern edges of Lake Okeechobee. Both sampling methods predicted ticks in narrow bands along the southern coast of the state and into the Florida Keys, although surveying was not performed there. The western panhandle showed substantial variation depending on survey methods (Figure 2) with convenience sampling indicating wider and more diffuse suitability.

Despite the broad geographic similarities of the ensemble maps (Figure 2), there was little agreement between the maps using the two sampling methods at a more detailed (hectare = pixel) spatial resolution (Table 3 and Table 4). 

Statistical comparison of survey methods: There was a statistically significant but only ‘slight’ agreement between ensemble SDMs using standard and convenience sampling for *A. americanum* (Kappa = 0.16; *p* < 0.05). Similarly, *I. scapularis* showed ‘slight’ agreement between standard and convenience sampling ensembles (Kappa = 0.12; *p* < 0.05). These results are evident in cross tabulations of the surveys showing extensive areal extents of model disagreements (Table 3 and Table 4). For example, although standard surveys excluded *A. americanum* from 46.6% of the mainland and convenience samples excluded questing ticks from 42.3% of the region (Table 2), only 48.6% (33,185 ha) were the same locations (Table 3). Conversely, only 13.8% of the entire state (1,415,116 ha) where all five of the models from the standard surveys indicated *A. americanum* would occur had the same prediction using the convenience samples. Models using convenience sampling excluded *A. americanum* from 34.5% of most likely regions of occurrence, based on standard survey SDMs (486,860 ha; Table 3). Similarly, for *I. scapularis*, only 12.9% of highest risk regions (1,382,180 ha) identified from standard survey SDMs were predicted using SDMs from convenience samples and 30.9% of most likely regions excluded *I. scapularis* using convenience SDMs (Table 4).

Sources of disagreement: Examining the survey data where models disagreed identified an unanticipated data quality issue. A total of 49/178 (27.5%) of convenience sampling sites that yielded *I. scapularis* were outside the distribution predicted from standard survey ensemble models (Appendix A). These were initially identified as errors of omission for the standard surveys. However, visiting these locations and further examination identified issues with the original data that were grouped into one of Jacquez’s [9] errors; sites immediately adjacent to at least one SDM, sites (within the software’s measurement error for the size of the raster cells), sites with obviously incorrect locations, sites with possible positional errors, and sites that may indicate limitations of the standard SDMs (true omission errors). There also was a statistical association between time intervals when convenience samples were collected and the proportion of their locations falling outside the standard SDMs (X^2^ = 13.14; 3 df; *p* = 0.004). Ticks collected from 1980, onwards were more likely to fall within the standard SDMs than ticks collected prior to 1980. However, there was no significant difference (X^2^ = 0.008; 1 df; *p* = 0.86) in the proportion of convenience samples after 1980, that were potential omission errors based on their data source (FLMNH vs. State of Florida).

For *I. scapularis*, 12 sites (6.7%) were within a pixel (immediately adjacent) to at least one SDM (Appendix A). Fourteen (7.9%) were geocoded to obviously incorrect sites (e.g., Tampa Bay, Chipola River; midlines of highways, street intersections, centers of buildings). Other likely misidentified geocoded tick locations were buildings on large properties, such as farms or ranches (n = 5; 2.8%), or likely positional errors, such as large, unbranched line segments (long stretches of roads), or large polygons with ‘poorly formed’ [9] location descriptions (n = 13; 7.3%). This resulted in five locations for *I. scapularis* (2.8%) that were likely errors of omission (Appendix A).

Among the 53 convenience sample sites for *A. americanum*, eight (15.1%) were potential errors of omission for the standard ensemble results. Only one (1.9%) was immediately adjacent to at least one standard SDM. The remaining seven sites were reported to the State of Florida and reported with specific, locatable addresses, eliminating obviously incorrect sites, although standard positional errors or misreported collection sites remained possible.

## 4. Discussion

SDMs and resulting ensemble models represent powerful analytical tools to represent the distributions of vectors and their pathogens over broad geographic areas [2,6,14] that do not rely on arbitrary geopolitical boundaries. The models provide an important approach to focus attention for investigations of disease outbreaks, implicating hosts or sources of vector borne and zoonotic agents and targeting public education and intervention. As such, the accuracy of these models is critical for public health decision making. To date, there has been extensive development of methods to generate ensemble SDMs but less attention to issues of study design, including how sampling strategies impact the subsequent results [6,40,41,42,43]. Consequently, maps presented as, de facto, correct are, instead, hypotheses that should be further evaluated for their predictive accuracies.

The CDC’s efforts to promulgate prospective standard survey designs for data collection and identifying what can be properly inferred from various sources of data is a critical step to generate high quality, comparable data throughout the United States for vectors of public health concern [1,2,3,4,5]. Despite this utility, it creates challenges for field survey teams by requiring substantial attention to planning and record keeping prior to initiating surveys [5]. Historical records and passive surveillance data are often obtained for other reasons and that makes it difficult for them to meet the standards for these surveys [44]. Passive data have utility in TBD studies, by finding TBD pathogens or vectors in a specific locale or region but they are tangential to the broader goals of SDMs of predicting the distributions of vectors where data have not been gathered or information is limited [7,45]. The ability of SDM software to generate results while blind to the quality of the input data represents a serious concern in generating accurate surveillance results.

A major, previously unresolved, question was whether standard surveys produced substantially different ensemble models than convenience samples when the difference was in the sampling methods [14,17]. This study examined the question in more detail. The results indicate that the two survey methods produce different results, especially at a spatial scale intended for planning surveys and interventions. However, at a coarser spatial scale the results are more ambiguous and suggest the appeal of convenience samples for a broad overview of vectors’ ranges. This study also identifies substantial limitations when using historical records, and indicates they should be used advisedly and with the anticipation that they may lack the needed spatio-temporal accuracy for many applications.

SDM comparisons: In addition to intended differences in the origins of the positive locations for ticks (convenience vs. standard surveys), the background sites for the convenience data were sampled randomly from the study area, rather than being surveyed and found to be ‘negative’ locations. As such, there is some concern that RF, LR, BRT and GAM datasets could include positive sites that ‘contaminate’ the comparisons. If we use the standard survey data to estimate the proportion of transects where ticks were absent in Florida, then 82.5% and 88.4% for *A. americanum* and *I. scapularis*, respectively sites would be negative. This is similar to other studies of tick distributions using SDMs e.g., [29]. Another potential cause for differences in the ensemble SDMs could be that the geographic/environmental ranges of the survey locations were drawn substantially different subregions so the collections sampled different subsets of the environment. Based on the collection sites (Appendix A–D), this seems unlikely as the entire mainland shows broad coverage by both survey methods.

The results of the SDMs used in the ensembles fell into two groups when classifying environmental identifiers of the tick species, BRT and RF, and to a lesser extent MARS, often selected the same environmental predictors, but weighted them differently. Additionally, these models were robust to the survey method and identified many of the same environmental factors, although, again, they were weighted differently. In contrast LR and MaxEnt selected many fewer environmental variables (Table 1). The identified variables also differed whether convenience or standard surveys were performed but within sampling schemes they tended to be similar (Table 1). The similar parameter selections for LR and MaxEnt might be anticipated as the methods are approximately equivalent [45]. The similarity in variable selection in BRT and RF is more surprising as these often are viewed as alternative algorithms. While these are both are machine learning approaches, they differ in how they build the classifiers, with RF bagging its randomly built trees [33] while BRT uses the error in the prediction of the model to weight the selection of variables for the subsequent trees [32].

Among the climatic variables, precipitation metrics were often associated with tick distributions (Table 1), and they tended to be identified regardless of sampling method or species. Given the association of microclimatic humidity and tick survival [46], their relationships are anticipated. Similarly, features of the environment, such as distance from surface water (distwater) and depth to local water table (dtwt) capture similar aspects of microhabitat. Elevation (dem), steepness of slope (curv) and measures in vegetation greenness (NDVI) also may reflect the rate of surface soil drying. Most of the temperature variables (bio 1–bio 11) were less often associated with tick occurrences (Table 1). Bio 9, the mean temperature during the driest quarter of the year, was the exception. The dry season, in Florida, occurs during winter months so that areas with the coldest climate in the state were most likely to have both species, and this occurred in the northern regions. This trend also is captured by bio 3, the tendency for daily temperature ranges to match the annual range of temperatures. Isothermality is associated with the southern portion of the state where neither species was commonly recorded.

Generally, models from standard surveys more efficiently identified where host seeking ticks were found than did ensemble models made from convenience surveys (Figure 1 and Figure 2). By the time all SDMs predicted occurrence (red regions), 70–80% of standard transects yielded ticks, while only 33–52% of sites in convenience sampling locations were sources of ticks.

Tick distributions in Florida: Geographic distributions, statewide, of the two species were broadly similar and confirmed historical surveys (reviewed in [17]). In the previous half century, *A. americanum* and *I. scapularis* were more likely to be the predominant questing species in the northern portions of the state, with decreasing numbers in central and southern areas [47,48,49,50]. In the central and southern regions *Amblyomma maculatum* tended to be more common while *A. americanum* was rarely, if ever found [50,51]. Allan and colleagues [50] surveying hunted white tailed deer (*Odocoileus virginianus*) and wild boar (*Sus scrofa*), noted that *A. americanum* prevalence decreased from 17.9% (n = 70) of hosts in the northern regions to 0% (n = 85) in the southern regions, on *O. virginianus,* and 5.2% on *S. scrofa* in the central (n = 19) to 1.5% (n = 24) in the south. These trends were also described in previous SDMs using standard surveys [22] and imply that choropleth maps using state or county boundaries [52,53] overestimate exposure by ignoring heterogeneity in local tick population dynamics. Alternatively, estimating local questing tick densities [3,4] provides improved characterization of exposures compared to crude occurrence estimates. Although density estimators also have challenges in interpretation, especially when seasonal population dynamics and averaging misconstrue longer term abundances [54].

Geographically, standard and convenience survey methods had their greatest divergence in predictions in the northeastern region of the state, where standard surveys identified areas along the Atlantic coast into the central ridge as most likely to yield ticks, while the convenience sampling moved this region further westward toward the Gulf coast. In a practical sense, this difference is important as the human population density, as well as outdoor recreational activity along the northeastern coastal region is elevated compared to the northwest. The two survey methods also indicated different distributions along the mid Gulf coast area with broad, but low, likelihood from central Florida to the coast for convenience sampling. The standard surveys restricted both tick species, but especially *I. scapularis*, along the Lake Wales Ridge (Figure 2A,C), a highland running north-south through the spine of central Florida to the Everglades. The ridge is associated with upland pine-oak forests and sandy scrub vegetation [55]. Convenience sampling indicated a broader, more diffuse distribution in the central part of the state that was not tied to obvious geographic/habitat features.

SDMs for both survey methods identified the two species in local environments in a narrow band along the southern coast from Fort Myers south and eastwards towards Miami and through the Florida Keys (Figure 2A–D). In this region, the standard surveys never yielded any ticks by flagging [7,17] but environmental factors were similar enough to those in the remaining portion of the state to predict their occurrence (Figure 2A,C; Table 1). That standard surveys generated ensemble maps predicting both tick species in regions where repeated sampling failed to yield questing ticks suggests that common environmental features along the southern coast capture critical features needed to maintain vector populations. Because of climatic differences between north and south Florida, we anticipate that the key environmental factors are landscape and vegetation, rather than climatic, features. It also supports suggestions by others that *I. scapularis* in southern Florida search for hosts differently from the more northerly populations that quest on vegetation and surfaces of leaf litter [46].

Convenience sampling did document ticks in this region, as had previous studies from vertebrates [49,50,51]. These were predominantly immatures removed from lizards in museum collections [15]. This speaks to the greatest strengths of convenience sampling—that using alternative sources and methods of sampling overcomes the limitations of any one single sampling strategy [56].

Sources of SDM discrepancies: This study identified two aspects of analyzing vector distributions using convenience sampling with SDMs (or other approaches to spatial modelling) that appear overlooked in applied risk analysis. The first, sampling bias, has been discussed as related to these tick species [7] and more generally in building SDMs [40,42,43,57,58]. Biased sampling with SDMs, as with other observational epidemiologic studies, risks generating inaccurate conclusions when attempting to generalize from the locations of the sample to a broader universe [43]. This occurs regardless of how well the models appear parameterized (internal validity). Standard designs serve to control or reduce the impact these biases [3,4].

We focus on a second challenge for retrospective, convenience sampling that has been overlooked. SDMs build their analyses by linking the geographic locations of tick collections with the environmental conditions at those sites. However, if the collection sites are incorrectly aligned with the environmental layers the incorrect values of the environmental data may be associated with the presumed tick locations. The extent of this problem is likely to be influenced by both the spatial resolution of the data and the extent of spatial autocorrelation in the environment.

Historically, locations and descriptions of local environments were recorded in collectors’ field survey notes. These data gave other investigators important clues when searching for ticks in other regions [59]. Current applications in computer sciences, remote sensing and geographical sciences have pushed the development of geographic information systems [10] and the growth of various, digital environmental databases. This simplifies linking collection sites with a vast array of environmental data, but at the spatial accuracy of the data, evaluating temporal changes in the environment are rarely reported.

The issue of positional error of health data [9] has been explored in medical and spatial geography for data sets most frequently when street addresses were used to locate health outcomes [11,12,13]. There are various reasons positional errors arise with these data sources [9]. Spatial positioning errors can arise from the software algorithms that translate an address to geographic coordinates, even when correct, specific street addresses are used because algorithms make assumptions about the positioning of addresses along street segments [9,10]. Because street segments are often longer in rural areas, these errors tend to be larger [9]. Address positional errors have received special attention in exposure studies for rural populations [11,12,13]. In these studies, house addresses had reported positional errors, with 44–72% with errors of at least 100 m and the average positional errors ranged from 200 to >800 m (with extreme values of 35.6–48.5 km). Positional errors by geocoding services, occurred even to the level of census tracts (2–15% of locations [12]). Address errors also may arise as input errors. Individuals may report a home address or the address from which a tick was sent to an archive, rather than the address from where it was collected (e.g., a state park or recreation area). Such errors are difficult to trace but need to be considered as part of data acquisition.

Location positioning that relies on surrogate geographic features, rather than numbered street addresses (e.g., centroids of property parcels) also can generate geographic errors. When street numbers are not available, geographic landmarks (e.g., 8 km south of Myakka River State Park) or place names (e.g., Alachua County) can be used. These locators may not be recorded to sufficient spatial resolution (is it 8 km, 8.0 km, 8.00 km?; where within Alachua County?) to meet the needs of the analysis. Without initial evaluation, the GIS software may geocode these locations with no indication of the measurement uncertainty.

Another potential source of attribution error arises with historical data when there is a temporal mismatch in the outcome data and the extracted environmental predictors [7,17]. In our study, most of the convenience samples were obtained during the 1990s—relatively close to the time when much of the environmental data were generated. However, convenience samples acquired prior to 1980 were more likely to be flagged as omission errors in the SDMs of the standard maps than were more recent records. The causes for this are not entirely obvious. Examining imagery of the convenience sites (Appendix A) for habitat characteristics rarely showed recent landscape modifications suggesting alterations of the habitat. Field surveys to the sites identified only three convenience sites that showed obvious evidence of recent anthropogenic modification (housing/road construction). The majority of locations appeared undisturbed within the recent time period

In summary, while convenience samples may be the ‘best available data’ for generating anticipated vector ranges and SDMs will ingest the data, the analyses do not produce the same results as those generated from a designed sampling framework. While they may provide coarse, sweeping, characterizations of vector ranges that are generally similar to SDMs from standard surveys, they should not be assumed as accurate for detailed decision making. Such outcomes should not be surprising as even the most basic scientific training recognizes that haphazard data aggregation does not produce reliable results and such data make it difficult to generalize the observations from samples to the population, at large.

Spatial modeling would grow, as a field, if it considered the results of studies to be hypotheses of the underlying phenomenon, such as vector distributions, more generally. It would raise basic research questions such as what factors limit the distribution of these human biting ticks to a portion of the region and whether it represents a non-equlibrial condition (the vector ranges are increasing or decreasing). Within this framework, extensive collaboration among field biologists, physiologists and those trained in observational epidemiologic studies, study design and geographic information sciences would form a productive team so we would be more certain of phenomena needing further study without having to suspect that the results were artifacts of the data themselves.

## Figures and Tables

**Figure 1 diseases-10-00032-f001:**
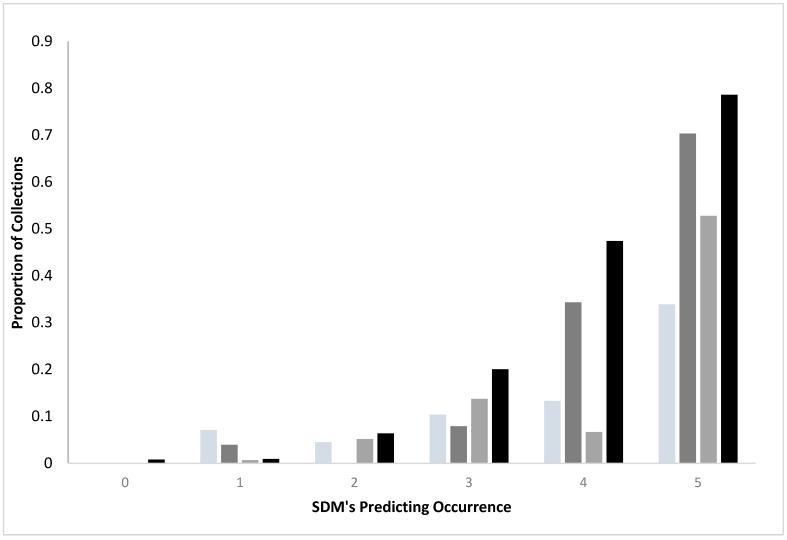
Proportions of collection locations yielding *I. scapularis* or *A. americanum* as the number of SDMs identifying tick occurrence increased. Standard surveys used transects without ticks to generate the denominators. Convenience samples used randomized background locations to generate the denominators. 


*I scapularis*, convenience (n = 1958 locations); 
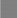

*I. scapularis*, standard (n = 560 locations); 


*A. americanum*, convenience (n = 583 locations); 


*A. americanum* standard (n = 560 locations). Standard surveys were more likely to yield ticks than did convenience sampling when there was concordance (SDM ≥ 3) among SDM’s (X^2^ = 53.60, 11 df, *p* < 0.0001 *A. americanum*; X^2^ = 326.25, 11 df, *p* < 0.0001 *I. scapularis*).

**Figure 2 diseases-10-00032-f002:**
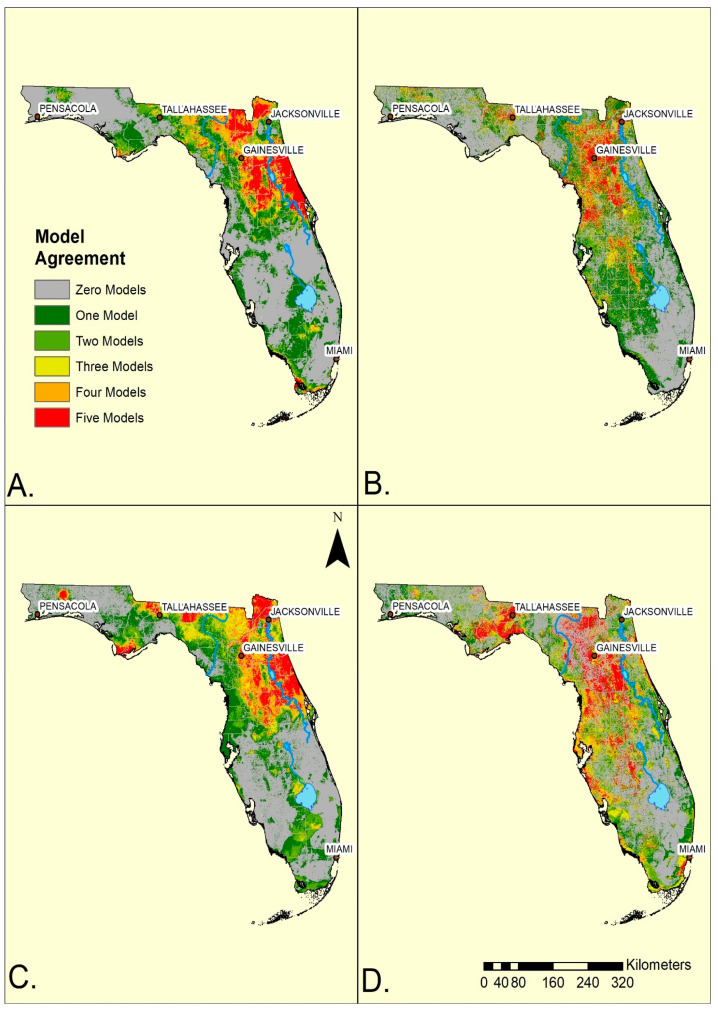
Ensemble models of *A. americanum* occurrence within Florida using standard survey (**A**), and *A. americanum* occurrence using convenience sampling (**B**). Ensemble models of *I. scapularis* occurrence within Florida using standard survey (**C**) and *I. scapularis* using convenience sampling (**D**). There was only slight agreement of ensemble maps, as measured by Kappa, for *A. americanum*, (Kappa = 0.016; Maps (**A**,**B**)) and for *I scapularis* (Kappa = 0.012; Maps (**C**,**D**)).

**Table 1 diseases-10-00032-t001:** Environmental variables selected for SDMs (Logistic, BRT MARS, RF, and MAXENT) for Convenience Sampling (Convenience; variables in italics) and Standard Survey (Standard; variables in bold) methods. *Amblyomma americanum* (top) and *Ixodes scapularis* (bottom).

*A. americanum*									
*Logistic*		*BRT*		*MARS*		*RF*		*MaxEnt*	
*Convenience*	Standard	*Convenience*	Standard	*Convenience*	Standard	*Convenience*	Standard	*Convenience*	Standard
*bio3*	**bio3**	*bio3*	**bio3**	*bio3*	**bio3**	*bio3*	**bio3**	*bio3*	
*bio4*		*bio4*		*bio4*		*bio4*			
	**bio9**	*bio9*	**bio9**			*bio9*	**bio9**		**bio9**
			**bio11**				**bio11**		
	**bio12**	*bio12*	**bio12**	*bio12*	**bio12**	*bio12*	**bio12**		**bio12**
	**bio13**		**bio13**		**bio13**		**bio13**		
		*bio14*	**bio14**		**bio14**	*bio14*	**bio14**		
			**bio15**				**bio15**		
		*bio18*				*bio18*			
*bio19*		*bio19*				*bio19*			
*dem*	**dem**	*dem*	**dem**	*dem*	**dem**	*dem*	**dem**		
	**curv**	*curv*	**curv**			*curv*	**curv**		
*ndvimin*		*ndvimin*	**ndvimin**	*ndvimin*	**ndvimin**	*ndvimin*	**ndvimin**		
	**nvimean**		**ndvimean**		**ndvimean**				**ndvimean**
*ndvimax*		*ndvimax*				*ndvimax*			
	**distwater**	*distwater*	**distwater**		**distwater**	*distwater*	**distwater**		**distwater**
		*grass*	**grass**	*grass*		*grass*	**grass**	*grass*	
		*forest*	**forest**	*forest*		*forest*	**forest**		
		*wetlands*	**wetlands**	*wetlands*		*wetlands*	**wetlands**	*wetlands*	
	**shrub**		**shrub**	*shrub*		*shrub*	**shrub**		
		*detwt*	**detwt**			*detwt*	**detwt**		
** *I. scapularis* **									
** *Logistic* **		** *BRT* **		** *MARS* **		** *RF* **		** *MaxEnt* **	
** *Convenience* **	**Standard**	** *Convenience* **	**Standard**	** *Convenience* **	**Standard**	** *Convenience* **	**Standard**	** *Convenience* **	**Standard**
		*bio3*		*bio3*		*bio3*			
		*bio4*		*bio4*		*bio4*			
	**bio6**		**bio6**		**bio6**		**bio6**		**bio6**
							**bio7**		
			**bio8**		**bio8**		**bio8**		
*bio9*		*bio9*	**bio9**	*bio9*		*bio9*	**bio9**	*bio9*	
	**bio12**	*bio12*	**bio12**	*bio12*	**bio12**	*bio12*	**bio12**		**bio12**
		*bio13*	**bio13**	*bio13*	**bio13**	*bio13*	**bio13**		**bio13**
*bio14*		*bio14*	**bio14**	*bio14*		*bio14*	**bio14**	*bio14*	
		*bio17*	**bio17**	*bio17*		*bio17*			
		*dem*	**dem**	*dem*		*dem*	**dem**		
	**curv**	*curv*	**curv**	*curv*	**curv**	*curv*	**curv**		
		*ndvimin*	**NDVImin**	*ndvimin*		*ndvimin*	**NDVImin**		
			**NDVImean**				**NDVImean**		
		*ndvimax*		*ndvimax*		*ndvimax*			
	**distwater**	*distwater*	**distwater**	*distwater*		*distwater*	**distwater**		**distwater**
*grass*		*grass*	**grass**	*grass*		*grass*	**grass**	*grass*	
		*forest*	**forest**	*forest*		*forest*	**forest**		
*wetlands*	**wetlands**	*wetlands*	**wetlands**	*wetlands*		*wetlands*	**wetlands**	*wetlands*	
	**shrub**	*shrub*	**shrub**	*shrub*		*shrub*	**shrub**		**shrub**
		*detwt*	**detwt**	*detwt*		*detwt*	**detwt**		

**Table 2 diseases-10-00032-t002:** Total Suitable Area (sq km) by SDM agreement within the ensemble results for standard surveys (Standard) and convenience samples (Convenience). Numbers in brackets are the percentage of the mainland incorporated by the numbers of SDMs. Ensemble = 0 indicates that none of the SDMs predicted occurrence, while Ensemble = 5 indicates that all the SDMs predicted occurrence.

Ensemble	Standard *I. scapularis*	Convenience *I. scapularis*	Standard *A. americanum*	Convenience *A. americanum*
0	58,013 (39.5)	61,104 (41.6)	68,320 (46.6)	62,014 (42.3)
1	29,542 (20.1)	21,441 (14.6)	31,125 (21.2)	40,238 (27.4)
2	18,919 (12.9)	19,154 (13.1)	14,065 (9.6)	15,313 (10.4)
3	13,179 (9.0)	17,216 (11.7)	8710 (5.9)	12,195 (8.3)
4	13,279 (9.0)	11,303 (7.7)	10,384 (7.1)	7579 (5.2)
5	13,822 (9.4)	16,537 (11.2)	14,151 (9.6)	9415 (6.4)

**Table 3 diseases-10-00032-t003:** *A. americanum* SDM agreement between standard and convenience samples measured in sq km. Bolded cells are regions where the numbers of SDMs for standard and convenience surveys were in agreement. Off main diagonal elements showed discordance in classification based on survey strategy.

Agreement				Standard Survey				
		0	1	2	3	4	5	TOTAL
	**0**	**33,184.85**	1339.649	5594.45	2437.46	2532.10	4868.60	62,013.95
	**1**	20,311.57	**8898.76**	3606.80	1890.10	2451.88	3078.75	40,237.86
Convenience	**2**	6249.33	3184.98	**1488.14**	1010.98	1517.70	1862.10	15,313.23
Survey	**3**	4534.92	2758.37	1235.27	**978.22**	1232.57	1455.91	12,195.26
	**4**	2350.61	1608.98	1028.64	748.37	**907.00**	935.63	7579.23
	**5**	1688.32	1277.32	1111.63	1644.53	1743.12	**1950.17**	9415.09
TOTAL		68,319.60	31,124.90	14,064.93	8709.66	10,384.37	14,151.16	146,754.62

**Table 4 diseases-10-00032-t004:** *I. scapularis* SDM agreement between standard and convenience samples measured in sq km. Bold cells are regions where the numbers of SDMs for standard and convenience samples were in agreement. Off main diagonal elements showed discordance in classification based on survey strategy.

Agreement				Standard Survey				
		0	1	2	3	4	5	TOTAL
	0	**29,944.83**	10,627.14	6931.97	5068.29	4267.06	4264.80	61,104.09
	1	8927.07	**4404.14**	2854.61	1456.21	1716.07	2082.93	21,441.03
Convenience	2	7788.58	3826.18	**2267.14**	1522.36	1583.34	2166.29	19,153.89
Survey	3	6195.48	4259.18	2184.84	**1165.39**	1372.35	2038.91	17,216.15
	4	2634.33	2888.80	1771.81	1140.65	**1385.46**	1481.51	11,302.56
	5	2522.66	3536.64	2909.03	2826.40	2954.81	**1787.36**	16,536.90
TOTAL		58,012.95	29,542.08	18,919.40	13,179.30	13,279.09	13,821.80	146,754.62

## Data Availability

Not applicable.

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
