# Peer review of "Ensemble Models for Tick Vectors: Standard Surveys Compared with Convenience Samples"

_diseases, 2022, doi:10.3390/diseases10020032_

Round 1

Reviewer 1 Report

Generally, I liked this paper and think it presents useful information on the limitations of samples and data used in species distribution modelling. I think it will be suitable for publication after the authors address a few issues.

  1. Suggest you define early in the paper examples of the standard surveys (do you primarily mean flagging, small mammal trapping, or both) and convenience samples (including museum, published, and online records). Provide the readers with a good definition of each so that we don’t have to hunt down other publications.
  2. When you describe the standard tick surveys, you refer to flagging mainly. Is this the case (where the flag is waved in front of the observed as they walk along a transect) or were drags (where the flannel is towed behind the observer as they walk a transect) used?
  3. I am somewhat bothered by the way background locations were used as absences for the convenience data. I can see the logic behind this, and it was used in citation 29 and 36 (looks to me like these are the same reference listed twice in the references cited section) but in their case, they had reasonable confidence that the species they were modelling was not in the >2000 counties they used as background/absences. I am not sure you can have the same degree of confidence for the backgrounds used as absences in the convenience part of the current study. Selection of the background points can affect model performance, so it might be a good idea to randomly select 4 or 5 sets of backgrounds and fit each model using the different sets of background points. If the models perform the same, or similar, with each of the different background sets, then the analysis would be more convincing.
  4. Line 112 – this citation only deals with A. americanum; suggest you cite 14 instead.
  5. Line 128 and elsewhere – it seems to me the reference numbering breaks down here and that references 29 through 40 listed in the text do not refer to references 29 through 40 in the references cited section.
  6. Line 130 to 140 – Do you reference the software used to estimate each of these models? Was it the dismo package? You should explicitly state that here.
  7. I also noticed that even though you did K-fold validation, you do not present any of the measures for model fit one might normally see, such as an ROC curve, AUC, etc. Also, there is really no detail given on model selection results for the convenience samples.
  8. I would like to see a map of the locations of positives for each sampling method. We don’t really get any visual sense of where these were located throughout the state.

Reviewer 2 Report

A comparison a SDM results based on ensemble simulations (5 models) are used to compare different sampling data and two disease vectors (two tick species). The paper focuses on comparing standard sampling methods and “convenience” samples.

The topic is interesting but the reading is not made easy by confusing expressions such as line 272 “sites immediately adjacent to at least one SDM” and lots more before that.

The paper would benefit from structuring more the text with sub-headings and more figures  and tables including maps of the sites. It is difficult to follow as it is.

A diagrammatic representation of the methodology would help to follow the different steps in the comparison(s). There is long discussion without subheadings which would help following through … and no conclusion?

If convenience samples means non-standardised protocol used to acquire data, perhaps this should be at specified in the abstract. In the same sort of details, the kappa for agreement needs to be a bit more specific, locations? spatial distribution for a given grid / scale? … or simply across the tick species?

Line 24 -25 “…geocoding algorithm failing to correctly locate convenience samples ..” ? explain why? is it a raster  based SDM versus vector data  vector issue? (be careful of not confusing people with disease vectors (ticks) and vector data (spatial points).

Line 49 perhaps here convenience samples could be defined

Line 72-73 difficult to understand what’s going on.  “ the standard sampling 72 generated ensemble models of SDMs that were consistent“ needs further clarifications. Do you mean the standard sampling made the ensemble simulations using SDMs more consistent (i.e. less variations)? or do you mean (following sentence) introduce a temporal consistency?

Line 87-103 a table describing properly the samples characteristics would help … and here the maps of the sample sites too.

Line 106 vs line 110 I suppose here you used existing SDMs and din’t create them, or cerated species distributions from running SDMs on samples.

Line 137 I was wondering if using both convenience and standard sampling could be used for ground truth when assessing predictive accuracies? (i.e. in the cross validation systematically adding the other sample data when predicting)

Line 144 is “suitable” the right term for this?  how about likelihood threshold?

Line 164 a problem in this sentence!

Figure 1 may calls for further analysis of this type? I ma not sure if I understand well Figure 1, is it the global map proportion for example where piwels/sites were identified for tick by 5 SDM or is it the the average of the local proportions (and in that case sd would be welcome)?

In the discussion, it would be interesting to discuss the fact that SDM are working on the basis of a relatively homogeneous response to environmental variables. Therefore, in some ways if the coverage of the sample matches the coverage of the variations of the predictors one could expect  a better accuracy in prediction? Could that be explanation that standard protocol makes the SMDs simulations more geographically accurate? (line 238)

Line 256 I am wondering what would be the agreements between the different samples would be better if considering only scores greater than 2, i.e. when already there are relatively good agreements between the SDMs?

Line 270 table S1 should be included as map in the article idem with the standard sample locations

Line 271 272 you mean as declared “ suitable” from at least one SDM …  

Line 279 (line 176) masp showing the sites could help here!

Line 291 “SDM agreement between standard and convenience samples measured in sq km” is missleading and should be

     “SDM agreements from … and from … for the xx sites / 146754.62 km2”

Line 434 Wouldn’t be the case to say they don’t have the same purpose? one giving regular “homogeneous” estimates (from SDM) and one raising alarms of potential changes?

Round 2

Reviewer 1 Report

The authors addressed most of my concerns in the second revision. My major concern was whether or not the background samples could confidently be considered as absence locations for the convenience data modelling. The authors provided a pretty good justification of this, and the added point maps help clarify the matter. I think the authors' discussion on this point should be included in the paper. They could start off with a brief description that several of the modelling methods require some confidence in the absence (background) portion of the data (MaxEnt does not, hence the reason it was developed) and that this may be lacking in studies using convenience sampling. They could then provide the information on why they have this confidence in the background data used for developing models from their convenience sampling data.  This would amount to condensing their response to a short paragraph and locating it near the beginning of the discussion. This would be a relatively minor edit and I would not need to see the paper after it is done. 

Author Response

We have incorporated Reviewer #1 comment into the Discussion section lines 361-371.

Reviewer 2 Report

The authors have brought some answers but without sometimes being much clearer and bringing other confusions. I think the papers misses still quite a lot of clear explanations in order to be useful to readers. The changes from previous version are not enough. Altogether, I am not recommending the paper to be published without considerable substantial improvments.
